# Smallholders' Livelihood Resilience in the Dryland Area of the Yellow River Basin in China from the Perspective of the Family Life Cycle: Based on GeoDetector and LMG Metric Model

Xueping Li and Xingmin Shi *

School of Geography and Tourism, Shaanxi Normal University, Xi'an 710119, China
* Correspondence: realsimon@163.com; Tel.: +86-13186068586

**Abstract:** Farm households' sustainable livelihoods in the dryland area of the Yellow River basin is an important guarantee of ecological protection and high-quality development for the Yellow River basin. However, farm households in this region have been facing frequent droughts, water resource shortages, severe soil erosion and other problems; their livelihood security has been seriously threatened. This study used a livelihood resilience framework to evaluate farm households' livelihood resilience in dryland areas through the field survey data and identified the influencing factors of livelihood resilience using the GeoDetector and the Lindeman, Merenda and Gold method (LMG) from the family life cycle perspective. The results revealed the following points: (1) there were significant differences in livelihood resilience, adaptive capacity and anticipatory capacity at each stage of the family life cycle at a 5% significant level. (2) The top two variables of livelihood resilience were preparedness and planning, and substitutable assets, followed by household characteristics. With the evolution of the family life cycle, the impacts of family assets and basic service access on livelihood resilience showed a "U" trend. On the contrary, savings and safety nets showed an inverted "U" shape. (3) Both the GeoDetector and LMG metric models could identify the key influencing factors, but the variable importance rankings of the two models were different to some degree. Finally, based on the results of the analysis, this study proposed targeted policy recommendations for building livelihood resilience of farm households.

**Keywords:** livelihood resilience; family life cycle; GeoDetector; LMG metric model; dryland area; China

## 1. Introduction

The concept of "sustainable livelihoods" was proposed during the 1980s to the early 1990s [1], which mainly comprises the following five components: capabilities, assets, activities, resilience and natural resources [2]. Among them, livelihood resilience in the face of stresses and shocks is central to both livelihood adaptation and coping [3]. Most scholars believed that this concept not only involved people's livelihood vulnerability but also their ability to resist external disturbances and their adaptability after a disturbance [4,5]. Livelihood resilience aims to sustainably manage resources for human development and well-being from the micro-level of smallholders [6], which has become a powerful tool to explore sustainable livelihoods [7,8]. Since resilience is characterized by stability, dynamics and steady-state transformation [9], integrating resilience into livelihoods is conducive to understanding how smallholders cope with adverse external disturbances and how they stabilize their livelihoods, as well as understanding livelihood system dynamic properties [10]. Building resilient livelihoods means that livelihood strategies of specific households are better able to cope with diverse impacts caused by adverse shocks and high uncertainties, manage livelihood risks and adapt to changing conditions [11] to promote sustainable livelihood development.

The notion of resilience was first introduced into ecosystem studies by Holling [12], which was defined as the ability of an ecosystem to absorb disturbances and maintain

its function and state [9]. Afterwards, resilience was gradually applied to more complex system sciences, such as social-ecological systems. With the gradual deepening of research concerning resilience to the social-ecological system, scholars gradually focused on the concept and theoretical study of livelihood resilience [13], which broadens the research field of resilience [14].

Livelihood resilience research mainly includes the following three aspects: the theory research of livelihood resilience, livelihood resilience assessment and analysis based on several analytical frameworks and exploring influencing factors of livelihood resilience.

(1) Livelihood is identified as a way of making a living that individuals, families or groups depend on [8], which is composed of capabilities, assets and activities [2], focusing on the connection between assets owned by people and livelihood decisions, except for income [15]. Due to the vulnerability of their living environment, the livelihoods of highly poor and disadvantaged groups are hampered, which leads them to be unable to escape the development dilemma. The sustainable livelihood approach is a powerful livelihood tool, which takes livelihood capitals as the core and underscores transforming livelihood capitals and activities into livelihood outcomes. This method is expected to find the entry point to formulate future development strategies to strengthen the development capacity of these vulnerable groups and help them gradually remove livelihood obstacles to achieve livelihood sustainability [15]. The notion of livelihood resilience, proposed as a component of sustainable livelihoods [2], inherits the research paradigm of resilience to social-ecological systems and highlights the role of human agency and our individual and collective capacity to cope with stressors [13]. For example, people can use social networks to solve the issues of information asymmetry and resource shortages and improve the ability to cope with disasters. Resilience is an inherent attribute of a system based on the adaptive cycle theory of social-ecological systems [16]; thus, study on livelihood resilience is a further improvement of the study on livelihood system mechanisms. Livelihood resilience places people at the center to solve livelihood development needs and the limitations of livelihood activities of the poorest and most vulnerable groups with the resilience theory and provides opportunities for them to achieve sustainable livelihoods. In terms of the concept of livelihood resilience, some scholars proposed the concept of livelihood resilience by combining the concepts of livelihood and resilience, but there no consensus has been reached. The current concept that is widely applied was proposed by Tanner et al. [13], which emphasizes the capacity of all generations to maintain and improve livelihood opportunities and human well-being in the face of external disturbances.

(2) Livelihood resilience assessment is the primary task for managers to understand people's ability to cope with disturbance and formulate resilient management strategies. At present, the comprehensive evaluation method through the analysis framework is more popular. Therefore, it becomes important to build an appropriate analysis framework. Several researchers have put forward a variety of analytical frameworks according to their own research needs, but there is still no universal analytical framework. Currently, the popular analytical frameworks are the framework of buffering, self-organization and learning capacity proposed by Speranza et al. [17] and the resilience index measurement approach (RIMA), proposed by the Food and Agriculture Organization [18], respectively. In addition, Sina et al. [8] developed a measuring livelihood resilience framework, comprising individual livelihood coping capacity, individual well-being, access to livelihood resources and social-physical robustness of local community, applying thematic analysis based on a literature review. Bahadur et al. [14] developed a 3As (adaptive capacity, anticipatory capacity and absorptive capacity) framework in the project "Building resilience and adaptation to climate extremes and disasters project", funded by the Department for International Development (DFID), which constructed a comprehensive indicator system at household and community levels. The assessment facilitates the integration of liveli-

hood and resilience, as well as issues related to human agency and empowerment, which is in line with the notion of livelihood resilience proposed by Tanner et al. [13]. Because of this, this study used this framework to assess the livelihood resilience of smallholders.

Secondly, as livelihood resilience aims to solve the problem of sustainable livelihoods of the poorest and most vulnerable groups, vulnerable ecological areas, poverty-stricken areas and disaster-prone areas are the focus of scholars. The dryland area, lying in the south of the Yellow River basin, is one of the areas with a vulnerable ecological environment in China, where the agricultural production has been faced with frequent droughts, serious soil erosion, and water shortage for a long time [19,20]. Therefore, the livelihood security of smallholders in this area is seriously threatened. Meanwhile, smallholders' livelihoods in this region have been deeply affected by social and economic transformations. For example, due to the impact of COVID-19, farmers' agricultural production and off-farm activities have been severely restricted, and their incomes have significantly decreased, which increases the vulnerability of their livelihoods. Therefore, building resilient, sustainable livelihood needs to be paid enough attention by managers. In 2019, China's central government put forward a strategy for ecological protection and high-quality development in the Yellow River basin, taking the issue of people's livelihood sustainability in this region to a new height. However, studies on the livelihood resilience of smallholders in the dryland area of China are very limited. It is necessary to evaluate the livelihood resilience of farmers in the dry tableland area, which aims to provide a theoretical reference for relevant government departments and managers to understand the livelihood resilience level and establish a resilient management system for the study area.

(3) Identifying the key factors of livelihood resilience can help managers to integrate existing resources and formulate effective livelihood resilience improvement policies for vulnerable groups. Some scholars have attempted to explore the influencing factors of livelihood resilience. For instance, Ado et al. [21] studied influencing factors of farmers' resilience to food security in the Aguie district of Niger and showed that family size, agricultural production and agricultural experience were the most important determinants of farmers' resilience to food security. Li et al. [22] revealed that education investment, social network, family burden ratio, and family size were the major factors that influenced the livelihood resilience of relocated migrants in China. Wen et al. [23] revealed that savings, per capita income, educational investment, educational level of household head, as well as social networks, were the core factors that influenced the resilience of households on the Loess Plateau in China. These pieces of literature have directly or indirectly confirmed the significant impacts of household characteristics and related factors on livelihood resilience, which are closely related to the family life cycle. The concept of family life cycle (FLC) was first proposed by Rowntree [24], which refers to the process of birth, development and disappearance of a family [25]. At present, the impact of FLC on smallholders has attracted the extensive attention of scholars. The existing literature mainly focused on the relationships between FLC and farmland management scale [26–28], smallholders' farmland transfer behaviors [29], livelihood strategy [29,30], rural labor transfer [31], and multidimensional poverty [32,33]. They explicitly or implicitly reflected that there were differences in capital accumulation, family size, dependency burden, livelihood risks and livelihood strategies of a household as the evolution of FLC [28,34], which lead to different livelihood resilience levels. However, there are few related studies. Therefore, this study attempts to explore the influencing factors of smallholders' livelihood resilience from the perspective of the FLC, which helps stakeholders to understand the research mechanism of livelihood resilience, and provide references for local governments and managers to formulate targeted resilient livelihood measures.

Additionally, in terms of research techniques, statistical methods, such as the structural equation model [35] and regression model [36], are more popular. However, some important influencing factors will be excluded from the analysis because of some overlap

between the influencing factors and indicators system, which is usually ignored in an estimation. Therefore, it is necessary to explore the relative importance of influencing factors to the evaluated index of livelihood resilience. In recent years, GeoDetector has been widely used to explore the influencing factors of spatial heterogeneity, but this method can also measure the explanatory powers of factors to dependent variables using statistical data [37]. Meanwhile, the Lindeman, Merenda and Gold (LMG) method is a method of relative importance assessment based on a linear regression model [38]. Therefore, both methods can be used to explore the important influencing factors of livelihood resilience.

Through the above literature review, this study mainly has three contributions to the existing literature, which are as follows: (1) we evaluated the livelihood resilience of smallholders in the dryland area of the Yellow River basin based on the 3As framework proposed by Bahadur et al. [14]. On the one hand, by combing the 3As framework and livelihood resilience concept accepted by most scholars, we believed that the 3As framework fully considered the role of human agency and empowerment, so it has certain advantages in livelihood resilience assessment. On the other hand, the dryland area has an important strategic position in the Yellow River basin of China; thus, building smallholders' livelihood resilience in this region is related to the implementation of a high-quality development strategy for the whole Yellow River basin. Considering this, we filled this gap to evaluate smallholders' livelihood resilience in the dryland area of the Yellow River basin in China. (2) Both GeoDetector and LMG metric models were used to identify the important factors that affect smallholders' livelihood resilience. The estimation of influencing factors of livelihood resilience is limited due to its widely inclusive indicator system. The above models can estimate the relative importance of the factors of livelihood resilience to identify crucial factors. (3) We provided a new perspective for analysis. FLC is a sociological concept, which this research introduced into the field of geography. Taking it as an entry point, we analyzed the differences in the livelihood resilience of smallholders in different stages of the FLC and their influencing factors, and aimed to help local governments formulate a targeted resilient livelihood management pathway to promote sustainable livelihood development in the dryland area of the Yellow River basin.

The remaining sections of this paper are organized as follows: the second section is the introduction of the study area. The third section is materials and methods, including data collection, theoretical framework and research methods. The fourth and the fifth parts are the results and discussions, respectively, which are the important contents of this study. The final part is divided into conclusions and suggestions.

## 2. Materials and Methods

### 2.1. Study Area

The dryland area is located in the south of the Yellow River basin in China (Figure 1), with a total population of about 18 million. The per capita of arable land in the study area is less than 0.133 ha. The dry land accounts for about 75% of the total arable land [39]. Therefore, the contradiction between the population and farmland is prominent in the study area. Topographically, a flat and open stretch slopes from north to south, with an altitude of 251~2779 m. The climate type of the study area can be described as a warm temperate, semi-arid continental and monsoon climate with four distinct seasons. Its average annual precipitation ranges from 400 mm to 650 mm, mainly concentrating in July to September every year. In addition, its annual average temperature is between 8.3 °C and 13.5 °C. The type of agriculture in the dryland area is rainfed agriculture that is identified as one of the important agricultural production areas in the Yellow River basin of China. Meanwhile, the study area is an important fruit and food production basis for the country, including high-quality grain, apple and pear [40]. Due to the vulnerable ecological environment and the climate-related impacts, the serious soil erosion, frequent flood and drought disasters, diseases and insect pests have been challenges for the sustainable agricultural development in the study area. In addition, with the continuous development of urbanization and industrialization, smallholders changed their farmland utilization

and livelihood strategies. Meanwhile, they are faced with diverse pressures, such as children's education and enrollment and cash gifts, which aggravated the vulnerability of their livelihoods.

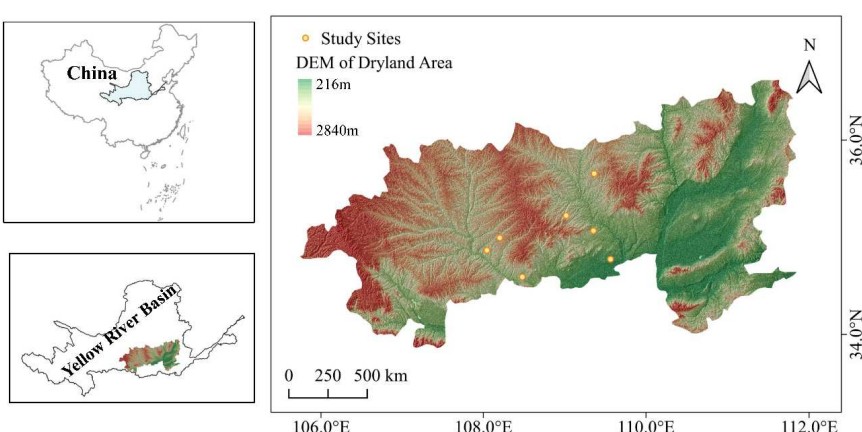

**Figure 1.** The study area location and the survey site distribution.

### 2.2. Data Collection

Data collection included the following two phases: pre-survey and formal survey. The presurvey was conducted in September 2019, which aimed to investigate the rural production and living conditions in the study area, such as farmers' livelihood types and strategies, and rural infrastructure construction. After obtaining the pre-survey results, finally, we revised and finalized the formal questionnaire. The formal survey was conducted in November 2019. The stratified random sampling method of "county-township-village-smallholders" was adopted in this phase to select the interviewed samples. Firstly, seven counties or cities were randomly selected from the study areas, which are shown in Figure 1. In the next step, we randomly chose one or two townships from each county or county-level county. Then, a total of eight administrative villages were determined. Finally, 40–45 smallholders were selected randomly from each administrative village as respondents. The face-to-face interviews to the household heads were implemented with structured questionnaires. Generally, the interviews lasted from 40 to 60 min. During the interview, the investigators introduced the investigation's purpose and explained any doubts for the respondents. Finally, a total of 353 questionnaires were distributed, of which there were 342 valid questionnaires, excluding the incomplete or error data, with an effective rate of 84.7%. Additionally, the focus group discussions for the village cadres were carried out to collect the agricultural production and rural basic situations.

The questionnaire involved a total of 43 questions, which were mainly divided into the following three parts: (1) basic information on a smallholder, such as household size, gender, ages, education levels and occupations of family members; (2) livelihood capital possessed by a smallholder, including natural capital, physical capital, financial capital, social capital and human capital; (3) livelihood strategies and stresses faced by a smallholder.

### 2.3. Theoretical Framework

The theoretical framework in the current study is shown in Figure 2. Livelihood resilience refers to "the ability of all people across generations to sustain and improve their livelihood opportunities and well-being despite environmental, economic, social and political disturbances" [13], which is mainly measured by the 3 As analysis framework in this study. The 3As analysis framework was proposed by Bahadur et al. [14] in the project "Building resilience and adaptation to climate extremes and disasters", funded by the Department for International Development (DFID), which initially aimed to help the communities in South and Southeast Asia, East Africa and the Sahel becoming more resilient to the climate-related shocks and stresses to ensure the vulnerable groups' well-being. Bahadur et al. [14] deconstructed resilience into three easily identifiable abilities,

adaptive capacity, anticipatory capacity and absorptive capacity, which interlinked each other. In particular, adaptive capacity refers to the ability of a social system to cope with long-term, ongoing future risks, and to learn and adjust to the adverse consequences, such as salinity. Anticipatory capacity is the ability of a social system to avoid or reduce the negative climate-related effects and extreme events through preparation and planning before the shocks and stresses. In addition, absorptive capacity refers to the ability of a social system to use skills and resources to cope with and manage adverse statuses, emergencies or disasters, such as hurricanes. Campbell [41] adopted this analysis framework to estimate the livelihood resilience of coffee growers living in Cedar Valley of Jamaica. According to the indicator system constructed by Bahadur et al. [14] and Campbell [41] and the actual situation and data availability in our study area, we constructed 25 indicators and 8 dimensions to evaluate smallholders' livelihood resilience. The specific indicator system is shown in Table 1.

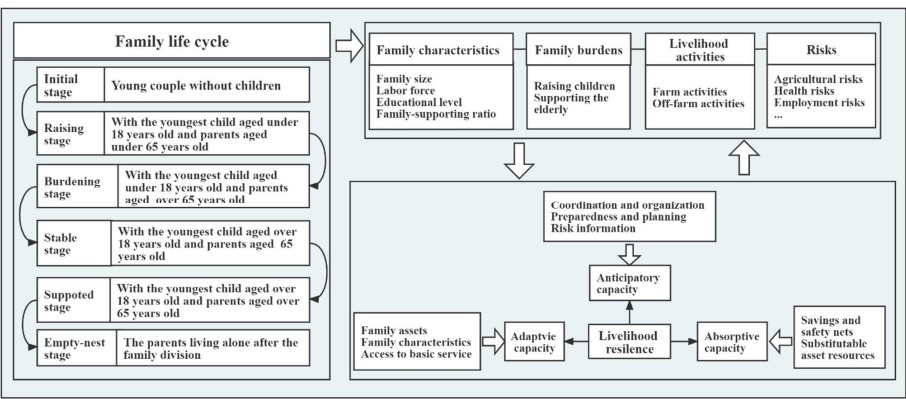

**Figure 2.** Theoretical framework of this study.

The concept of FLC was first proposed by Rowntree [24] to solve poverty issues, which refers to the repeated process from birth to the vanishing of a family [25]. Due to the cultural differences in different regions in the world, the criteria of the FLC division should fully consider family characteristics in the study areas. Given this, this study integrated the relevant studies on the Chinese FLC and referred to the work by Ye et al. [29]. Considering whether the children in a family were over 18 years old and whether the elderly were over 65 years old, the given smallholders in the dryland area of the Yellow River basin in China were divided into six types, which were the initial stage, raising stage, burdening stage, stable stage, supported stage and empty-nest stage, as shown in Figure 2.

FLC is a comprehensive reflection of human capital characteristics, which evolves with the changes in the household size and the quantity and quality of the labor force in a family [27,28]. The existing literature revealed the close relationships between the FLC and labor supply, household income, consumption and savings, and livelihood strategies [25,26,42,43]. For example, after the birth of a child in a family, the raising burdens of the family gradually increase. Then, the obligation to support the elderly appears as the FLC evolves, when the burdens of the family reach the maximum. As the children reach adulthood, the burden reduces, which reduces to the minimum when there are no people to support or raise. Some studies indicated that family income in different stages of the FLC was different, and the relationship between FLC and family income presented an inverted "U" shape. Furthermore, FLC had a significant impact on off-farm labor transfer. The research by Lin and Wang [44] revealed that the probability of off-farm labor transfer showed a trend of an inverted "U" shape with the evolution of the FLC. In addition, smallholders at different stages of the FLC faced different risks and shocks [34]. After reviewing the existing literature, it is found that not only the household characteristics and burdens, such as labor force and family size, but also the livelihood activities and risks faced by the smallholders, are closely related to their resilience to livelihood. Therefore, taking FLC as the entry point, it is crucial to explore the resilience of

smallholders in different stages to external pressures and shocks for scholars and managers to dynamically understand the resilience of farmers' livelihoods.

**Table 1.** The indicator system and variable weights of livelihood resilience assessment.

| Components | Dimensions | Indicators | Descriptions of Indicators | Roles | Weights |
|---|---|---|---|---|---|
| Adaptive capacity | Family assets X1 | Farmland areas A1 | 1 = 5 mu and below, 2 = 5~10 mu, 3 = 10~15 mu, 4= 15~20 mu, 5 = 20 mu and above (1 mu is about 0.067 ha) | + | 0.120 |
| | | Durable goods A2 | Number of durable goods possessed by a household/piece | + | 0.060 |
| | | Housing area A3 | The total area of housing owned by a household/m$^2$ | + | 0.120 |
| | Family characteristics X2 | Education level A4 | The educational level of a household head, 1 = primary school and below, 2 = junior high school, 3 = senior high school, 4 = junior college, 5 = university and above | + | 0.134 |
| | | Health A5 | The general health of family members, 1 = very unhealthy, 5 = very healthy | + | 0.134 |
| | | Family-supporting ratio A6 | The ratio of non-labor population to total population in a family/% | - | 0.270 |
| | Access to basic services X3 | Traffic convenience A7 | The smallholders' distance from the nearest market /km | - | 0.066 |
| | | Medical facility A8 | Supporting facilities of hospital nearest to a family, 1 = very few, 5 = many | + | 0.066 |
| | | Irrigation water A9 | Whether farmland can be continuously irrigated, 1 = yes, 0 = no | + | 0.200 |
| Anticipatory capacity | Coordination and organization X4 | Skill training B1 | Whether family members participated in farm skill training, 1 = yes, 0 = no | + | 0.149 |
| | | Farmer-benefiting policies and projects B2 | Whether there were farmer-benefiting policies and projects, 1 = yes, 0 = no | + | 0.149 |
| | Preparedness and planning X5 | Strategies to coping with agricultural disaster risks B3 | Number of measures adopted to deal with agricultural disaster risks (such as increasing pesticides, irrigations, plastic mulching, soil moisture conservation techniques, changing crop varieties and cropping structures) | + | 0.216 |
| | | Diversity of livelihood activities B4 | Whether a family adopts off-farm measures to resist external disturbance, 1 = yes, 0 = no | + | 0.323 |
| | Risk information X6 | Disaster risk information B5 | Number of ways to obtain disaster risk information, 1 = very few, 5 = many | + | 0.082 |
| | | Weather forecast B6 | Whether you often follow the weather forecast, 1 = yes, 0 = no | + | 0.082 |
| Absorptive capacity | Savings and security nets X7 | Savings C1 | 1 = CNY 10,000 and below, 2 = CNY 10,000–40,000, 3 = CNY 40,000–70,000, 4= CNY 70,000–100,000, 5 = CNY 100,000 and above | + | 0.157 |
| | | Agricultural insurance C2 | Whether a smallholder buys agricultural insurance, 1 = yes, 0 = no | + | 0.054 |
| | | Assistance from kith and kin C3 | 1 = very little, 5 = very much | + | 0.090 |
| | | Government assistance C4 | Whether you get assistance from local government, 1 = yes, 0 = no | + | 0.032 |
| | | Credit services C5 | Whether farmers have access to credit services, 1 = yes, 0 = no | + | 0.166 |
| | Substitutable asset resources X8 | Off-farm income C6 | Whether the family has off-farm income, 1 = yes, 0 = no | + | 0.300 |
| | | Crop diversity C7 | Number of species of crops | + | 0.200 |

*2.4. Methods*

2.4.1. Livelihood Resilience Measurement

In this paper, the analytic hierarchy process (AHP) and technique for order preference by similarity to an ideal solution (TOPSIS) were adopted to calculate adaptive capacity,

absorptive capacity and anticipatory capacity. Before calculating, we used the range normalization method to standardize the data.

According to the methods of decomposition, comparative judgement and comprehensive thinking, AHP regards the study respondents as a hierarchical system to quantify the factors from each hierarchy. Compared with the expert scoring method, this method integrates the quantitative method into the qualitative method to make the subjective scoring more reasonable. In addition, the objective weighting method, such as the entropy evaluation method, assigns weights according to the dispersion degree of data distribution. Specifically, when the values change greatly, the weights of these indicators assigned are greater. Therefore, this method easily ignores the importance of indicators. Comparatively, AHP has the advantage of considering the importance of indicators to the objectives. Hence, AHP was selected in this study to assign weights to the indicators, which all passed the consistency tests. The weights of the indicator system are shown in Table 1.

The TOPSIS method is a common multi-objective decision analysis method, which aims to evaluate the relative merits according to the distances between the positive and negative ideal solutions. The computation procedure is shown as follows:

① Building a dimensionless data matrix

$$(Y_{ij})_{m \times n} \tag{1}$$

② Computing the weight normalization matrix

$$(Z_j)_{m \times n} = (Y_{ij} \times w_j)_{m \times n} \tag{2}$$

③ Computing the positive ideal solution $Z^+$ and negative ideal solution $Z^-$

$$Z^+ = (Z_1^+, Z_2^+, \ldots, Z_n^+) = \{ max Z_{ij} | j = 1, 2, \ldots, n \} \tag{3}$$

$$Z^- = (Z_1^-, Z_2^-, \ldots, Z_n^-) = \{ min Z_{ij} | j = 1, 2, \ldots, n \} \tag{4}$$

④ Computing the Euclidean distance between each identified indicator and positive and negative ideal solutions

$$d_i^+ = \sqrt{\sum_{j=1}^n \left( Z_{ij} - Z_j^+ \right)^2} \tag{5}$$

$$d_i^- = \sqrt{\sum_{j=1}^n \left( Z_{ij} - Z_j^- \right)^2} \tag{6}$$

⑤ Computing the close-degree $C_i$ of the positive and negative ideal solutions and the surveyed respondents

$$C_i = \frac{d_i^-}{d_i^- + d_i^+} \tag{7}$$

where the values of $C_i$ are between 0 and 1. $C_i$ with a high value indicates that the evaluated objective is larger.

⑥ Computing the livelihood resilience $LR$

$$LR_i = \frac{AC_i + ABC_i + ANC_i}{3} \tag{8}$$

where $AC_i$, $ABC_i$ and $ANC_i$ represent the adaptive capacity, absorptive capacity and anticipatory capacity of the $i$-th sample, respectively.

### 2.4.2. The Specification of the GeoDetector Model

The GeoDetector is used to detect spatial heterogeneity of data and reveal its driving factors by a set of spatial variance analyses [38], which has been widely applied to study geography issues in recent years. It includes factor detector, interaction detector, risk detec-

tor and ecological detector. Among them, factor detection can measure the interpretation degree of the given indicators to livelihood resilience, which is measured by the q value [39], which is as follows:

$$q = 1 - \frac{1}{n\partial^2}\sum_{i=1}^{l} n_i \partial_i^2 \tag{9}$$

where $q$ indicates the explanatory power of the independent variable X to livelihood resilience, which ranges from 0 to 1. The larger the q value, the greater the power of X with regard to livelihood resilience. Furthermore, $n$ is the number of samples, $l$ is the number of involved indicators, $\partial^2$ and $\partial_i^2$ are the total variance and the variance in the $i$-th indicator, respectively.

### 2.4.3. The Specification of the LMG Metric Model

The Lindeman, Merenda and Gold method (LMG) decomposes $R^2$ into non-negative contribution based on the average orderings of explanatory variables from the multiple linear regression model [45], which is regarded as one of the parametric regression methods on the basis of variable decomposition. In this study, the LMG metric was conducted to measure the explanatory powers of the selected indicators to livelihood resilience. The specific calculation steps are shown as follows:

① Firstly, a multiple linear regression model (MLR) is established to calculate $R^2$, shown by the following equation:

$$R^2 = \frac{Model\ SS(model\ with\ variable\ in\ S)}{Total\ SS} \tag{10}$$

where *Model SS* represents the sum of squares of the model, and *Total SS* is the total sum of squares.

② When adding the regressors in a set $M$ to a model, the additional $R^2$ is defined as *seq $R^2$*, which is calculated by the following formula:

$$seqR^2\left(\frac{M}{S}\right) = R^2(M \cup S) - R^2(S) \tag{11}$$

③ The formula for $R^2$ allocated to the regressor $x_k$ in the order $r$ can be written as follows:

$$seqR^2(\{x_k\} \cup S_k(r)) = R^2(\{x_k\} \cup S_k(r)) - R^2(S_k(r)) \tag{12}$$

where $S_k(r)$ denotes a set of regressors entered into the model before regressors $x_k$ in the order $r$. The order of the regressors in any model is a permutation of the available regressors $x_1, \ldots, x_p$, which is expressed by the tuple of indicators $r = (r_1, \ldots, r_p)$.

④ The metric LMG for the regressors $x_k$ is calculated as follows:

$$LMG(x_k) = \frac{1}{p!}\sum_{r\ permutation} seqR^2(\{x_k\}|r) \tag{13}$$

The LMG metrics were conducted using the R package "Relaimpo" developed by Grömping [46].

## 3. Results

### 3.1. Description of Livelihood Resilience

The AHP-TOPSIS method was applied to calculate the three components of livelihood resilience, and then Formula (8) was used to calculate the livelihood resilience of smallholders in the dryland area in China.

### 3.1.1. Adaptive, Anticipatory and Absorptive Capacity

The results of one-way ANOVA showed (Table 2) that the adaptive capacity and anticipatory capacity of smallholders at different stages of the FLC were significantly different at the 5% and 1% levels, respectively. Specifically, all the average values of the adaptive, anticipatory and absorptive capacities of smallholders in the empty-nest stage were at their minimum, while the average values of the adaptive and absorptive capacity at the supported stage were the largest, as was the case for the average values of the expected ability of the initial stage.

**Table 2.** Mean values and one-way ANOVA of livelihood resilience and its components.

|  | Initial Stage | Raising Stage | Burdening Stage | Stable Stage | Supported Stage | Empty-Nest Stage | Mean Value | Homogeneity Variance | ANOVA |
|---|---|---|---|---|---|---|---|---|---|
| Adaptive capacity | 0.349 | 0.341 | 0.327 | 0.368 | 0.370 | 0.318 | 0.345 | 0.069 | 0.033 ** |
| Anticipatory capacity | 0.494 | 0.465 | 0.446 | 0.417 | 0.446 | 0.329 | 0.440 | 0.262 | 0.000 *** |
| Absorptive capacity | 0.298 | 0.284 | 0.286 | 0.280 | 0.320 | 0.262 | 0.287 | 0.687 | 0.597 |
| Livelihood resilience | 0.398 | 0.380 | 0.369 | 0.372 | 0.384 | 0.320 | 0.372 | 0.789 | 0.032 ** |

*** $p < 0.01$, ** $p < 0.05$.

Figure 3 shows the kernel density distribution of the adaptive, anticipatory and absorptive capacities. In terms of adaptive capacity, the curve of kernel density of the supported stage was relatively flat, indicating that data of that stage fluctuated greatly. The kernel density curves of the other stages were steep, especially in the initial stage, suggesting that the data distributions were concentrated. As for anticipatory capacity, the kernel density curve in the initial stage was flat, with values that largely fluctuated. The peak value in the empty-nest stage showed a left bias, which revealed that the anticipatory capacity of most families at this stage was small. In addition, the curve in the supported stage showed a steep trend, indicating that the anticipatory capacity of most families was concentrated in the mean value. As far as absorptive capacity was concerned, the kernel density curves of the supported stage and empty-nest stage showed a bimodal distribution, indicating that the absorptive capacity of farmers at this stage showed a polarization trend. The density curves of the other stages were steep with left-biased peaks, indicating that the absorptive capacity of most smallholders was small.

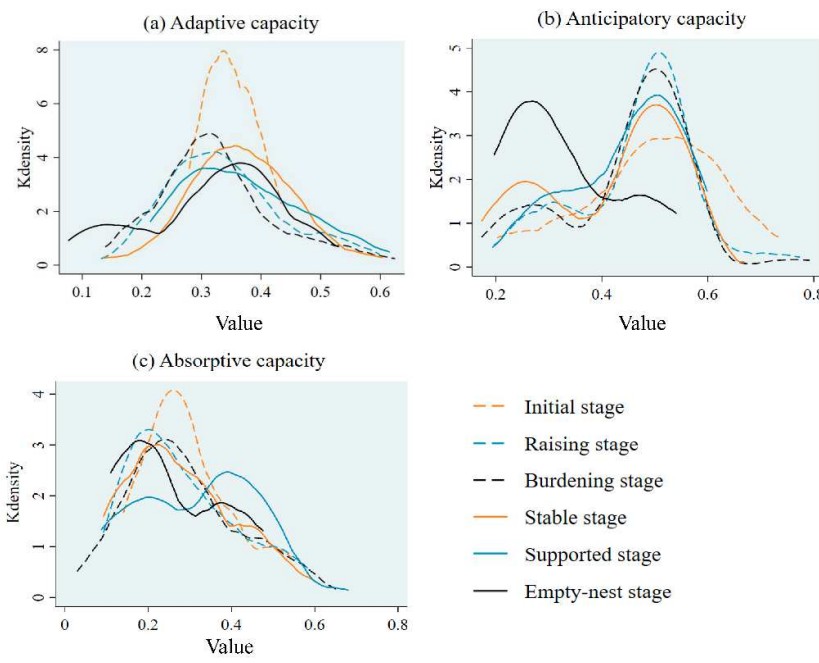

**Figure 3.** Kdensity of adaptive, anticipatory and absorptive capacity.

### 3.1.2. Livelihood Resilience

The results of one-way ANOVA also showed (Table 2) that the livelihood resilience of smallholders at different stages of the FLC was significantly different at the 5% level, which had an average value of 0.372, a maximum value in the initial stage and a minimum value in the empty-nest stage. Figure 4 showed the violin diagram of livelihood resilience in every stage. The box of livelihood resilience of farmers in the initial stage is close to the upper section, indicating that the livelihood resilience in this stage was high. The livelihood resilience in the empty-nest stage was narrow at the top and wide at the bottom, which revealed that the livelihood resilience in this stage showed low-value distribution. Livelihood resilience in the raising stage was close to the lower quartile, which showed a lower livelihood resilience in this stage. During the supported stage, on the contrary, the livelihood resilience was wide at the top and narrow at the bottom, indicating that the resilience of most farmers was high.

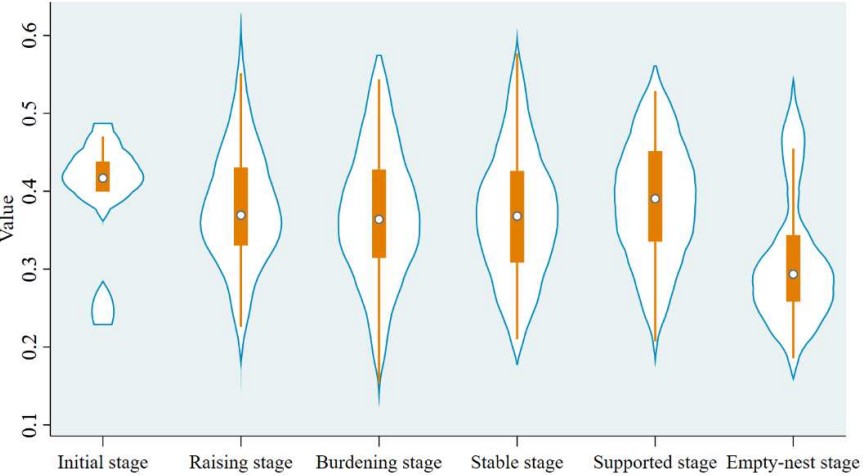

**Figure 4.** Violin plot of livelihood resilience in each stage of the family life cycle.

### 3.2. Influencing Factors of Livelihood Resilience

#### 3.2.1. The Results of Factor Detection

This study regarded the computed livelihood resilience index at each stage of the FLC as the dependent variable and took the dimensions of livelihood resilience as the independent variables. Then, the explanatory powers of the dimensions of livelihood resilience were calculated through Formula (9). Considering the small sample size in the initial stage of the study area and the fact that families in the initial stage will enter the raising stage soon after they build a family, this study incorporated samples in the initial stage into the raising stage, referring to the research by Ye et al. [29]. After that, according to the requirement of the GeoDetector model, the continuous variables among the independent variables were converted into discrete variables by the natural breaks method.

Factor detection revealed the explanatory powers of the influencing factors with regard to livelihood resilience at the dimension level. Table 3 showed the q values of eight dimensions in each stage of the FLC, which reflected that both preparedness and planning and substitutable asset resources were that most important variables in any stage of FLC. Meanwhile, coordination and organization, risk information, and access to basic service were not very important, especially coordination and organization. Additionally, family characteristics and family assets were the important variables in the raising stage. Family characteristics, family assets and savings and safety nets were relatively important factors in the burdening stage. According to the q values, the variable importance ranking of savings and safety nets was the third in the stable stage and supported stage, followed by family characteristics, access to basic service and family assets. As for the empty-nest stage, family assets and family characteristics were relatively important influencing factors.

**Table 3.** Explanation powers of eight dimensions contributing to livelihood resilience.

| | Family Assets | Family Characteristics | Access to Basic Services | Coordination and Organization | Preparedness and Planning | Risk Information | Savings and Safety Nets | Substitutable Asset Resources |
|---|---|---|---|---|---|---|---|---|
| Raising stage | 0.134 | 0.252 | 0.094 | 0.005 | 0.388 | 0.100 | 0.092 | 0.337 |
| Burdening stage | 0.195 | 0.187 | 0.052 | 0.054 | 0.383 | 0.083 | 0.132 | 0.425 |
| Stable stage | 0.145 | 0.219 | 0.160 | 0.100 | 0.380 | 0.065 | 0.273 | 0.453 |
| Supported stage | 0.108 | 0.221 | 0.174 | 0.052 | 0.343 | 0.045 | 0.264 | 0.505 |
| Empty-nest stage | 0.423 | 0.350 | 0.237 | 0.202 | 0.435 | 0.154 | 0.223 | 0.661 |

### 3.2.2. LMG Metric

To further explore the important factors of livelihood resilience, the LMG metric method was used to calculate the relative importance of eight dimensions. First, the variance inflation factor (VIF) analysis at the dimension level was conducted to test the identified regressors' multicollinearity. The results showed that the VIF of all regressors was less than 10 (Table 4), indicating that there was no multicollinearity between the regressors. Subsequently, multiple linear regression (MLR) was used to investigate the impacts of indicators on livelihood resilience from the perspective of the FLC. Finally, the LMG metrics of the factors in each stage were calculated based on the MLR model.

**Table 4.** The VIF analysis of eight dimensions contributing to livelihood resilience.

| | Family Assets | Family Characteristics | Access to Basic Services | Coordination and Organization | Preparedness and Planning | Risk Information | Savings and Safety Nets | Substitutable Asset Resources |
|---|---|---|---|---|---|---|---|---|
| VIF | 1.145 | 1.066 | 1.052 | 1.046 | 1.078 | 1.031 | 1.110 | 1.125 |

The variable importance rankings obtained by the LMG metrics are shown in Table 5. The results revealed that preparedness and planning, and substitutable asset resources were the most important influencing factors in all stages of FLC. Furthermore, both coordination and organization and risk information were not very important in any of the stages of FLC. As for the other dimensions, we found that family characteristics ranked third in the raising stage, followed by savings and safety nets, access to basic services and family assets. The q values of family assets and family characteristics were relatively large in the burdening stage. For the stable stage, the q value of family characteristics ranked third, followed by savings and safety nets, and family assets. In the supported stage, access to basic services, family assets, savings and safety nets, and family characteristics were relatively important variables to livelihood resilience. In the empty-nest stage, family assets, family characteristics and access to basic service were relatively important.

**Table 5.** The LMG metrics of eight dimensions' contribution to livelihood resilience.

| | Family Assets | Family Characteristics | Access to Basic Services | Coordination and Organization | Preparedness and Planning | Risk Information | Savings and Safety Nets | Substitutable Asset Resources |
|---|---|---|---|---|---|---|---|---|
| Raising stage | 0.076 | 0.134 | 0.078 | 0.004 | 0.335 | 0.003 | 0.079 | 0.291 |
| Burdening stage | 0.124 | 0.087 | 0.038 | 0.016 | 0.332 | 0.003 | 0.066 | 0.335 |
| Stable stage | 0.064 | 0.158 | 0.030 | 0.023 | 0.321 | 0.015 | 0.117 | 0.272 |
| Supported stage | 0.084 | 0.073 | 0.097 | 0.004 | 0.269 | 0.003 | 0.083 | 0.388 |
| Empty-nest stage | 0.192 | 0.141 | 0.074 | 0.012 | 0.249 | 0.054 | 0.049 | 0.229 |

### 3.3. Bivariate Correlations

In order to compare and analyze the result consistency of the GeoDetector model and LMG metric, we took the data of smallholders in the stable stage as an example to draw the bivariate correlations between livelihood resilience and the eight dimensions (Figure 5). The results showed that smallholders' livelihood resilience was strongly correlated with family assets (X1), family characteristics (X2), preparedness and planning (X5), savings and safety nets (X7) and substitutable asset resources (X8).

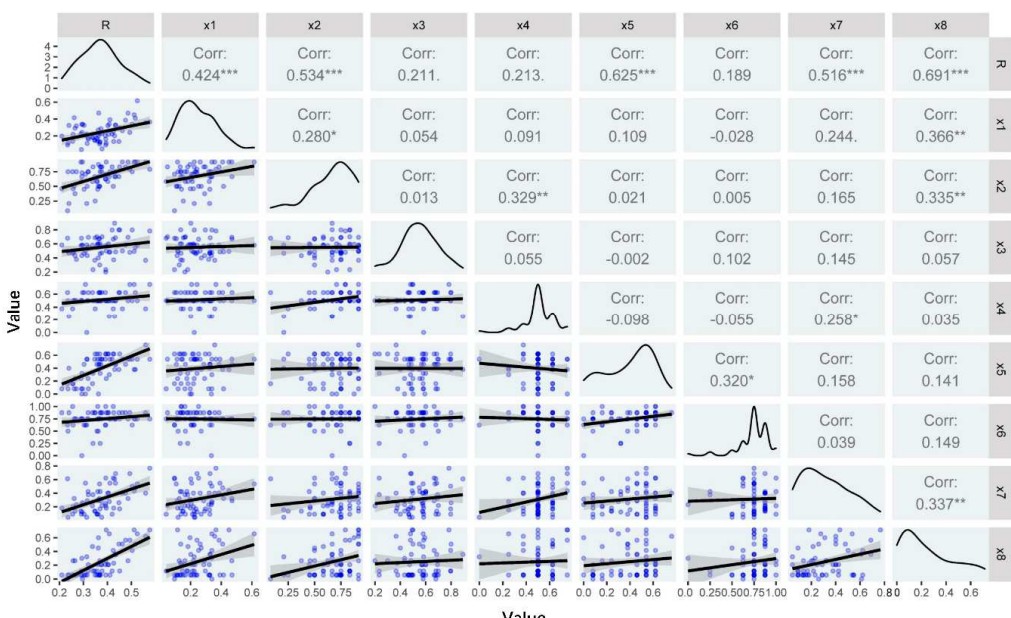

**Figure 5.** Bivariate correlations between livelihood resilience and eight dimensions in the stable stage.
*** $p < 0.001$, ** $p < 0.01$, * $p < 0.05$.

## 4. Discussion

### 4.1. Livelihood Resilience of Smallholders in Different Stages of the FLC

Through the analysis of livelihood resilience of smallholders, it can be concluded that livelihood resilience did not present a monotonically increasing or decreasing trend with the evolution of the FLC. The mean values of livelihood resilience in the burdening stage and empty-nest stage were lower than the mean value of all samples, while they were higher in the initial, raising, stable and supported stages, which might be explained by the fact that the dependency and support burdens of a family did not increase or decrease gradually with the evolution of the FLC, but varied in different stages. Families either in the initial and stable stages have no dependency or support burdens, while smallholders in the raising and supported stages have a single dependency or support burden. Particularly, smallholders in the burdening stage have both dependency and support burdens. Therefore, it can be observed that the livelihood resilience of smallholders in different stages is different. According to the field survey in the dryland area of China, it was common for the elderly in rural areas to take care of their grandchildren and perform farming duties, which increased the effective labor force to a certain extent [47] and reduced dependency and support burdens of a family. Hence, we found that this could effectively enhance the livelihood resilience of smallholders in the burdening and supported stages. Secondly, there were differences in social capital accumulation and depreciation possessed by smallholders in different stages of the FLC, which made information acquiring channels and social networks different. All of these ultimately led to the different levels of livelihood resilience of smallholders in different stages of the FLC.

Additionally, smallholders select different livelihood activities according to their needs and goals in the different stages of the FLC, which also result in different levels of livelihood resilience of smallholders. Previous studies showed that smallholders in the middle stages of the FLC were more likely to choose off-farm strategies than those at either end [33]; thus, those families have greater livelihood resilience due to higher off-farm income. In addition, with the evolution of the FLC, farm income of smallholders increased [28], as well as the agricultural risks they faced. Hence, the livelihood resilience of smallholders decreases. Therefore, it is important to conduct targeted agricultural skills training for smallholders in the burden and supported stages to enhance their ability to cope with agricultural risks and family income, which can promote smallholders' livelihood resilience.

Generally, members from smallholders in the empty-nest stage are more than 65 years old, so it is difficult for them to migrant working. More importantly, influenced by their physical health, farmers in the empty-nest stage generally tend to choose farming or retirement, which results in limited agricultural income and the minimum non-agricultural income for them. In addition, medical expenses are burdens for smallholders in the empty-nest stage. Therefore, poor family income and heavy medical expenses lead the families in this stage to have a poor ability to resist risks and vulnerable livelihoods. In other words, smallholders in the empty-nest stage have lower livelihood resilience. Given this, relevant managers should pay attention to the ability to access the medical and health of smallholders in the empty-nest stage and improve the living conditions of the elderly, such as building an elderly community and providing a variety of basic services in the community for them. Moreover, land transfer and a safety network in the empty-nest stage should be encouraged and built.

### 4.2. Analyzing the Influencing Factors of Livelihood Resilience

According to the factor detection results, preparedness and planning and substitutable asset resources had the greatest impacts on the livelihood resilience of smallholders at all stages of the FLC. Preparedness and planning were mainly measured by the farm strategies to deal with agricultural disasters and off-farm activities, which can reflect the ability of smallholders to cope with agricultural disasters and social shocks. Smallholders in the dryland area of China usually adopted soil moisture conservation technology, mulching, changing planting structure and crop varieties, increasing pesticides and irrigation to cope with agricultural disasters, as well as employing migrant workers and focusing on non-agricultural operation and other off-farm activities to resist climate change or social disturbances. Substitutable asset resources "can ensure that farmers can use other assets to stabilize their consumption and maintain health even if they suffer from a heavy asset loss caused by external disturbances" [14], which were measured by off-farm income and crop diversity. Secondly, family characteristics have a great impact on livelihood resilience. This may be because household characteristics (mainly including education levels, health status and family-supporting ratio) have a greater impact on the livelihood activities and capitals of smallholders in any stage of the FLC, which ultimately affects their livelihood resilience. Fan et al. [48] also confirmed that physical capital, human capital and social network have an important impact on farmers' livelihood vulnerability. Therefore, relevant managers should diversify farmers' livelihood strategies, and take targeted measures to improve their livelihood skills to improve income levels, such as agricultural and non-agricultural skill training in rural areas and the development of new agricultural management models.

This study also revealed that the explanatory power of family assets and access to basic services on livelihood resilience of smallholders showed a U-shaped trend, along with the evolution of the FLC. These were the crucial parts of adaptive capacity. On the contrary, the importance of savings and safety nets on the livelihood resilience of smallholders showed an inverted U-shaped trend with the evolution of the FLC. Specifically, family assets mainly involved farmland size, housing area and family durable goods, etc. Smallholders in the supported and burdening stages were built in the second round of the farmland contract period in China, so land consolidation and reallocation based on family member changes were not allowed, which resulted in these smallholders having limited farmland sizes [27]. However, this policy also made smallholders in these stages more likely to select off-farm employment, so that they generally possessed larger housing areas and more durable goods, and were more resilient to cope with the long-term livelihood pressures caused by external disturbances. In addition, they had fewer savings due to heavy dependency and support burdens. In order to improve the livelihood resilience of smallholders in this stage, relevant managers should conduct targeted off-farm employment skill training, encourage entrepreneurship, and provide policy support to smallholders. In addition, these families should be encouraged to transfer farmland to promote non-farm labor transfer. As for the smallholders in the supported stage, the importance of family assets to livelihood resilience

was small, while the importance of savings and safety nets was large, indicating that the impact of family assets on the livelihood resilience gradually decreased with the evolution of the FLC. Since smallholders in the burden and support stages tended to focus on farming due to family responsibilities, age, health and other reasons, agricultural skills training should be carried out for them to improve their agricultural income.

In terms of smallholders in the empty-nest stage, the impact of savings and safety nets on livelihood resilience decreased due to their low importance ascribed to it. Risk information and coordination and organization had less impact on livelihood resilience. In this study, risk information was mainly measured by disaster risk information and weather forecast, which could effectively prevent the adverse impact of extreme events on agriculture and reduce agricultural losses. The low importance of risk information may be caused by various information channels and low agricultural dependency; thus, the impact of risk information was small. Coordination and organization were measured by agricultural skill training and benefiting-farmers policies and projects. Smallholders in the dryland area in China reported that they received less farming skill training and expressed a lack of understanding of benefiting-farmer policies and projects. Given this, village cadres should promote farmer-benefitting policies and projects to ensure locals can fully enjoy preferential treatment.

### 4.3. Comparative Analysis

By comparing the results of the GeoDetector and the LMG metric, it was found that both methods could identify the important influencing factors of livelihood resilience. Specifically, the impacts of preparedness and planning and substitutable asset resources on livelihood resilience of smallholders in different stages of the FLC were similar in the two methods. Secondly, family characteristics had a great impact on livelihood resilience. Except in the burdening stage and empty-nest stage, the variable importance rankings of the two methods have poor consistency in the other stages. For instance, the bivariate scatter plots of smallholders' livelihood resilience and eight dimensions in the stable period proved that both the GeoDetector and LMG metric from the MLR model could identify the important influencing factors well, but there were differences in their variable importance rankings. This may be because the LMG metric based on the MLR model is a variance analysis method and decomposes $R^2$ into the regressors to calculate the relative importance of indicators, which can better identify the linear correlations between eight dimensions and livelihood resilience [38]. The GeoDetector, which uses a spatial variance analysis method, has no linear assumption on variables and is immune to multicollinearity among multiple independent variables [37], so it can also accurately identify linear correlations. To sum up, both methods can identify important influencing factors of livelihood resilience. However, the independent variables must be discrete in the GeoDetector model [37]. Therefore, according to the analysis requirement, we converted the independent variables into discrete variables before analysis. We think that this may be one of the reasons for the differences in the variable importance ranking of the two methods.

### 5. Conclusions

Sustainable livelihood development of smallholders in the dryland area is crucial for the implementation of the strategy of ecological protection and high-quality development in the Yellow River basin. For a long time, farmers in the dryland area of the Yellow River basin have suffered from natural and social shocks, which seriously threaten their sustainable livelihood development. Therefore, it is important to urgently analyze the resilience of farmers' livelihood in the dryland area. Given this, this study evaluated the livelihood resilience of smallholders in the dry land of the Yellow River basin based on the 3As framework, and then explored the relative importance of indicators of livelihood resilience from the perspective of the FLC. Finally, the target suggestions were put forward for local governments to build resilient livelihood management policies (Figure 6).

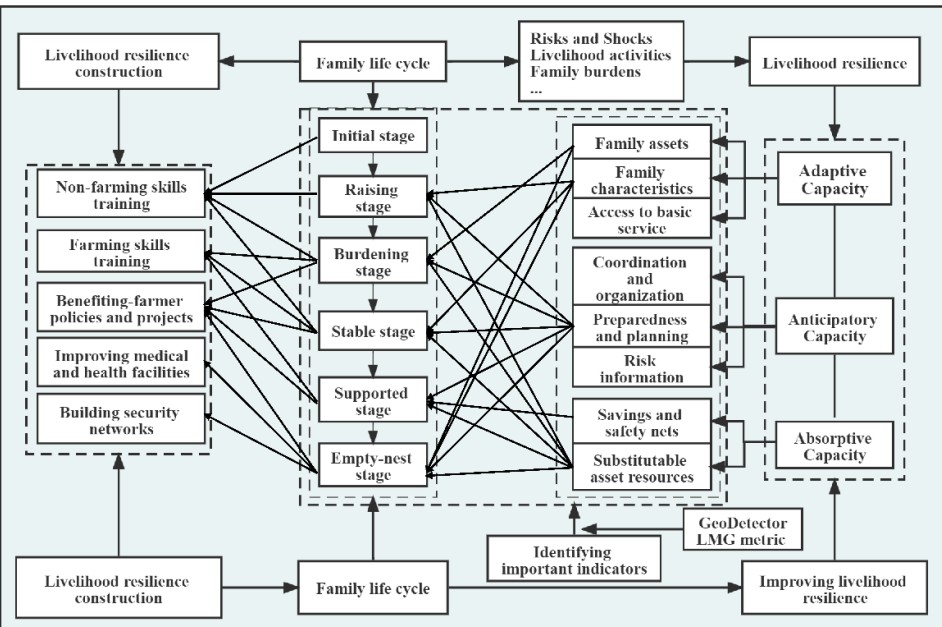

**Figure 6.** The mechanism of household livelihood resilience analysis and improvement from the family life cycle perspective.

According to the results and discussions, the conclusions were drawn as follows:

(1)    There were significant differences in the livelihood resilience, adaptive capacity and anticipatory capacity of smallholders in each stage of the FLC at a 5% level. Livelihood resilience in the burdening stage and the empty-nest stage was lower than the population mean value, while the resilience of livelihood in the initial stage, raising stage, stable stage and supported stage was higher than or equal to the population mean value.

(2)    Preparedness and planning and substitutable assets resources had the impacts on the livelihood resilience of smallholders at each stage of the FLC. And the factor of family characteristics also was an important determinant. Furthermore, the impact of family assets and access to basic services on the smallholders' livelihood resilience showed a U-shaped trend with the evolution of the FLC. On the contrary, the impact of savings and safety net on livelihood resilience presented an inverted U-shaped trend with the evolution of the FLC.

(3)    Comparing the results of the GeoDetector and LMG metric models, it was found that the two models can better explore the key influencing factors of livelihood resilience, such as family characteristics, preparedness and income, and substitutable asset resources. However, there were differences in the relative importance rankings of the given regressors in each stage of the FLC between the two models, especially in the raising stage and supported stage. Through comparison and analysis, we found that the results obtained by the GeoDetector were more accurate to explore the relative importance of indicators to livelihood resilience in this study.

Based on the results of this study and field survey in dryland areas, it found that preparedness and planning, substitutable asset resources, and family characteristics were identified as the important factors affecting smallholders' livelihood resilience. In addition, several indicators had different impacts on livelihood resilience at different stages of the FLC. Given this, when formulating the policies to achieve the livelihood sustainability of the locals, the government should give full consideration to the key influencing factors and the stages of the FLC for a smallholder. Specifically, the local government should carry out vocational skill training for farmers and promote labor force transfer in the first few stages to diversify family income, with the aim of building a resilient livelihood.

Secondly, managers should pay attention to farming skill training and the improvement of disaster prevention and mitigation capabilities for farmers in the later stages of the FLC to improve their livelihood resilience. In addition, diversified farm-benefiting policies and projects should be put forward to reduce farmers' agricultural input costs and living costs. Moreover, the ability to access basic services for smallholders in the empty-nest stage should be improved by improving basic medical and health conditions in the rural regions and living conditions of the elderly, encouraging families in the empty-nest stage to conduct farmland transfers to maintain their living standards and strengthen their ability to resist risks. Finally, local managers should focus on the safety network construction of smallholders in the empty-nest stage.

The problem of "empty-nest" discussed in this study is a social problem that has been widely focused on in China but is not a global social problem. For the studies on family life cycles in other regions, different criteria need to be considered according to the actual situations, so the results obtained in this study have a certain limitation. Additionally, this study explored the differences in livelihood resilience and the relative importance of its influencing factors from the perspective of the FLC, because smallholders in different stages of the FLC have distinctive structure characteristics. Future works need to examine the moderator role of FLC to enrich livelihood resilience research and provide suggestions for livelihood resilience construction in rural areas in the Yellow River basin of China.

**Author Contributions:** Conceptualization, X.S.; Methodology, X.L. and X.S.; Formal Analysis, X.L.; Writing—Original Draft Preparation, X.L.; Writing—Review and Editing, X.L. and X.S. All authors have read and agreed to the published version of the manuscript.

**Funding:** This study was supported by the Humanity and Social Science Youth Foundation of Ministry of Education of China (21YJA840014); the Shaanxi Provincial Key Research and Development Program (2021ZDLSF05-02); the Shaanxi Social Science Foundation Project (2020F004); the Shaanxi Provincial Basic Research Program (2022JM-15); and the Doctor Candidate Free Exploration Project of Shaanxi Normal University (2021TS017).

**Institutional Review Board Statement:** Ethical review and approval were waived for this study because data and information collected through face-to-face interview. Moreover, personal opinions on the research topics were treated in an aggregated form.

**Informed Consent Statement:** Informed consent was obtained from all subjects involved in the study.

**Data Availability Statement:** The data presented in this study are available upon request from the corresponding author.

**Acknowledgments:** The authors would like to express thanks to the editorial team of this journal. Song Z., Chen X.Y., Chen N., Cao Z.X., Qin Y.H., Wang X.X., Song J.Y., Wu S.J., Chen H.Q., Gao X.Y. and Zhao F. made contributions to the questionnaire survey.

**Conflicts of Interest:** The authors declare no conflict of interest.

**Research Involving Human Subjects:** This study was approved by the review board of Shaanxi Normal University and conducted in accordance with the approved relevant guidelines and regulations. All subjects gave their informed consent for inclusion before they participated in the study.

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
