# Peer review of "Smallholders’ Livelihood Resilience in the Dryland Area of the Yellow River Basin in China from the Perspective of the Family Life Cycle: Based on GeoDetector and LMG Metric Model"

_land, doi:10.3390/land11091427_

Round 1
Reviewer 1 Report
This is an interesting and meaningful study. Based on the survey data of rural households in arid areas, the authors discussed the characteristics of household livelihood resilience under different household life cycles from the perspective of household life cycle. In general, the research topic is novel, the research design is relatively reasonable, and the conclusions are basically in line with common sense, which can provide decision-making reference for regional development related policy making. It is suggested to be published after minor revision. The following points are for your reference:
(1) The introduction suggests a modest rewrite. Since the research starts from the perspective of family life cycle, it is suggested that the authors should pay more attention to the family life cycle. It now feels as if the author is talking more about the resilience of farmers' livelihoods and less about the family life cycle. Regarding the household cycle, the following article can be used as the author's reference:
Relationships between land management scale and livelihood strategy selection of rural households in China from the perspective of family life cycle
(2) The data presentation suggests more detail. For example, what are the selection criteria for sample counties, towns, villages and farmers? The length of each questionnaire, etc.
(3) In the introduction of the research framework, it is suggested that the core concepts involved in this research should be introduced as a basis, which can help readers to better understand the full text.
(4) Compared with the conventional methods, the author needs to further explain the uniqueness of the GeoDetector and LMG metric model.
Reviewer 2 Report
Dear Authors,
I found your article very interesting, I appreciated the analysis of the literature, the description of the methodology and in general the analysis that you have done. However, I would suggest to change something in the presentation of the results, for example I would change the comment of the results by avoiding placing the order of the dimensions for each stage and making a more general comment by highlighting some aspects and/or the most relevant differences, because the rest is inferred from the tables. I appreciated the presentation of the study area, but I would include the paragraph in the materials and methods and you could provide some more information on agriculture and farming practiced by small farmers.
Please find below some specific suggestions.
· Delete “and” between the authors names
· Bibliographic citations are done in the classic way (Names of authors and year) and not as required by MDPI (number in square brackets which is then reported in the bibliography) so please modify them
· I would generally recommend reviewing English, for example: Line 37 I think you should write: we need to resist “to” external….; Line 82 the sentence “no consensus to be reached” is not clear to me, maybe “to be” is not necessary; Line 100 I think that after “framework” you have to put "that is proposed”; Line 129 perhaps is “to integrate”, or in Line 147 I should put “to understand”; Line 173 is matric or matrix or metric?; Line 208 perhaps is “survey sites”, since there are 7; Line 236 to become or becoming; Line 326 after the point goes “Among” and not “among”; Line 395 “The box” and not “the box”; Line 480 I would say “we found that”; Line 499 I would say “to resist at risk”; Line 508 I think is “is usually adopted”; Line 511 “to resist to climate”; Line 568 is “The first” and not “the first”; Line 612 “we found that…”; Line 632 “to resist to risk”
· Line 86 putting the point (2) after all this time in my opinion only creates confusion, I would put the 3 aspects included in the research at the beginning, in line 57, and then I would explain them one by one. So, for example, in this case I would stick directly with line 87
· Lines 233-234 you should write “project” either before or in the title, I think it's missing here
· In table 1 I would better put the header text (on a single line) and put borders on the cells (grid type) because now the table seems a bit confusing to me
· Line 252 give a line of space between table 1 and the text
· Line 333 there is a sign in Chinese which I believe stands for “and”
· Line 335 R2 stands for R2?
· Figure 2 please enlarge a few boxes to make the text read better, the same for figure 6
· Line 373 I would put here table 2 and after the description of figure 3, the space occupied is the same but it makes more sense to put the description of figure 3 immediately before the figure
· Line 375 but what does relatively “gentle” mean?
· In table 2, 3, 4, 5 please adapt the writings in the header
· Lines 453 and 459 “Substitutable asset resources” has smaller font sizes and it seems to me that this happens in other places as well, so please check the font size
· Line 581 to 584: I have already seen the sentence before, I think it should be slightly modified
· Line 642 I believe that bold should be used only for “Funding” and not for the entire period, the same goes for the “Acknowledgments” and between the two it seems to me there is too much space
· Lines 466-468 you should try to better present it or somehow justify this statement because it is contrary to what you initially hypothesized on the basis of the analyzed literature
Reviewer 3 Report
"Smallholders’ livelihood resilience in the dryland area of the 2 Yellow River basin in China from the perspective of the family life cycle: Based on GeoDetector and LMG metric model "
The research takes up the relatively rarely discussed problem of smallholders' livelihood resilience in the conditions of unfavorable conditions for the functioning of family farms or (more broadly) families in agricultural areas. The article seems very interesting for a potential reader, especially in the aspect of vanishing climatic conditions, which means that the described problem may become more and more relevant.
The introduction to the research is extensive and contains all the key information. The citation method is probably not compliant with the journal's requirements, but it is a technical issue and can be corrected at the stages of preparing the article for publication after possible approval.
Data: The number of obtained questionnaires seems to be sufficient to perform the planned research.
Where do the weight values come from (last column of table 1)? If they come from other studies, are these values universal enough that they can be applied unchanged for other areas?
Does the used FLC concept ignore many modern families? The assumption that the beginning of families (also in agricultural areas) is always a young couple, perhaps still corresponds to the traditional agricultural areas of China (and not only China). However, you should be aware that this is subject to change. Has an analysis been carried out which proportion of the surveyed families does not correspond to any of the categories of the FLC model?
The last category (older parents live alone) as the last stage of family development is also puzzling. I don't know what it is like in China, but in many rural societies in the world this stage practically does not occur (extended families). The problem of "empty nest", important for the described research, is not always common, which should be referred to at the discussion stage, at the stage of assessing the universality of the research results obtained and the possibility of using them in a wider than the Chinese perspective. Although certainly due to the size and diversity of China also in this respect, the universality of the obtained results is certainly limited.
Methodology and results:
The methods used (AHP, Geodetector model, LMG metric model) were described in a sufficiently precise manner, and the results obtained using them are legible (although, for example, the axis description in Figure 3 is missing, this remark also applies to Figures 4 and 5)
The current Chapter 5 (Discussions) should probably be called: 5. Discussion.
In terms of content, its content is correct, but Figure 5 (Bivariate correlations between livelihood resilience and eight dimensions in the stable stage.) Should definitely be moved to the results section with a few sentences describing its content. At the discussion stage, we most often use the results, but it does not seem correct to enter this type of table with data in this chapter,
6.2. Suggestions. It seems to me that it is not necessary to create a separate chapter with this name. The current content may be part of chapter “6. Conclusions "(after changing it from its current name: "Conclusions and Suggestions "
Reviewer 4 Report
The paper presents interesting results about the resilience of smallholders in dryland of the Yellow River.
The Introduction shows a comprehensive review of previous studies that portrait the scientific contribution of the paper.
The Material and Methods are well described, as well as the Results.
The Discussion part can be improved by better underlining the main policy implications of the results.
